# Reviewing Research Trends—A Scientometric Approach Using Gunshot Residue (GSR) Literature as an Example

**Catarina Sobreira**[ID]**, Joyce K. Klu, Christian Cole**[ID]**, Niamh Nic Daéid and Hervé Ménard** *[ID]

Leverhulme Research Centre for Forensic Science, University of Dundee, Dundee DD1 4HN, UK;
c.sobreira@dundee.ac.uk (C.S.); j.k.klu@dundee.ac.uk (J.K.K.); c.cole@dundee.ac.uk (C.C.);
n.nicdaeid@dundee.ac.uk (N.N.D.)
* Correspondence: h.menard@dundee.ac.uk

**Abstract:** The ability to manage, distil and disseminate the significant amount of information that is available from published literature is fast becoming a core and critical skill across all research domains, including that of forensic science. In this study, a simplified scientometric approach has been applied to available literature on gunshot residue (GSR) as a test evidence type aiming to evaluate publication trends and explore the interconnectivity between authors. A total of 731 publications were retrieved using the search engine 'Scopus' and come from 1589 known authors, of whom 401 contributed to more than one research output on this subject. Out of the total number of publications, only 35 (4.8%) were found to be Open Access (OA). The Compound Annual Growth Rate (CAGR) for years 2006 and 2016 reveals a much higher growth in publications relating to GSR (8.0%) than the benchmark annual growth rate of 3.9%. The distribution of a broad spectrum of keywords generated from the publications confirms a historical trend, in particular regarding the use of analytical techniques, in the study of gunshot residue. The results inform how relevant information extracted from a bibliometric search can be used to explore, analyse and define new research areas.

**Keywords:** gunshot residue; bibliometric analysis; publication trends; database; Scopus

## 1. Introduction

The application of a range of scientific methodologies to questions of relevance to the investigation of alleged criminal acts is the core operational motivation behind the complex set of disciplines that make forensic science imperative for the administration of justice [1]. Since 2009 in particular, there have however been several reports, research papers and practical criticisms of forensic science and its applications. These have demanded change in all aspects of process and culture within the domain of forensic science, and we would advocate that this must also include attitudes, practices and access to research and research outputs.

The 2009 National Academy of Forensic Science (NAS) report: Strengthening Forensic Science in the United States: A Path Forward; the 2016 US President's Council of Advisors on Science and Technology (PCAST) report: Forensic Science in Criminal Courts: Ensuring Scientific Validity of Feature-Comparison Methods, and more recently the 2019 UK House of Lords Select Committee on Science and Technology report: Forensic Science and the Criminal Justice System: A Blueprint for Change all point to systemic failures in regard to validity, robustness and reliability of forensic scientific methods and how this negatively impacts the interpretation of evidence taken to court [2–4].

Open science has been suggested as one approach to address the trends of 'bad science' often encountered in a wide variety of domains of practice (such as medicine or psychology [5,6]), and we

would forcibly advocate that such trends are also widespread across areas of the forensic science literature. One of the goals of conducting research is to share the results obtained so that other researchers and domain experts can be aware of the findings and their impact on a specific research field. By not sharing findings and research methodologies, a verification of the results obtained can be challenging, raising questions around repeatability, reproducibility and accuracy of the data and the scientific standards that should be met. Forensic science practitioners rely on the knowledge researchers produce in their decision making when evaluating and interpreting evidence. In circumstances where the data generated through research are not made available, then they cannot be scrutinised, and any errors, misinterpretations, overinterpretations or erroneous results otherwise unrecognised might lead to unreliable evidence being used within the evidential process. A better way of undertaking the scientific research happens when, together with publications, related datasets are openly provided, and where further interpretation is welcome. Such an Open Access process is increasingly embraced across research domains and forensic science should be no different [7].

Bibliometrics is a statistical and quantitative method used to analyse publications and visualise networks associated with a body of literature (e.g., [8,9]). Such an approach is being employed in systematic reviews and can potentially reveal unknown relations between different areas of research, and such findings have been demonstrated in literature-based discovery (e.g., [10–12]) or the study of citation impact indicator (e.g., [13]). In this work, a simplified scientometric approach (a subfield of bibliometrics with a focus on analysing the scientific literature) has been applied to the available literature on gunshot residue (GSR) to identify patterns and trends in, for example, keyword, co-authorship, collaborative networks, country of origin and citations rates.

## 2. Materials and Methods

### 2.1. Publication Lists.

The list of records was compiled using Scopus. The following query was made "`TITLE-ABS-KEY (gunshot AND residue) AND PUBYEAR < 2019`". This search generated a list of output with gunshot and residue appearing in the record title, abstract or keyword list. No distinction was made regarding the document type (i.e., Article, Conference paper, Review, Book chapter, etc.). The exported list of records was then directly processed in R, an open-source software environment for statistical computing and graphics. The number of citations per output was also used as part of the analysis. Prior to analysis, when necessary, the Keywords list was checked for spelling mistakes and corrected to be more consistent for techniques and acronyms. A similar approach was also followed for the Authors lists. All the corrections were carried out in R. The research data underpinning this publication can be accessed at http://doi.org/10.5281/zenodo.3582799 [14].

### 2.2. Keywords Analysis.

For each output, when available, the keywords from the Authors and the journal index were combined, generating 4313 distinct keywords. The generated list was corrected to take into consideration variation in technique names, spelling mistakes, plural and/or formatting issues. A total of 447 distinct keywords (10.4% of the original list) were corrected. The amended list of keywords was finally attached to the original data set for further analysis. The frequency count per year of all the Keywords was compiled.

### 2.3. Authors.

As per the Keywords, the list of authors was amended to take into account missing initials and/or spelling mistakes including the presence or absence of special characters. The Author ID in the original data set was also used to reduce and validate the correction. If more than one Author ID was observed for one name, other parameters such as co-author(s), institutions, etc. were used to minimise the chance of establishing an incorrect author interconnection.

## 3. Results

### 3.1. Overview of Data

A total of 731 publications were retrieved following a Scopus search using the keywords "TITLE-ABS-KEY (gunshot AND residue) AND PUBYEAR < 2019". The majority of the 731 outputs were research articles (606), 58 were conference papers and 31 were review articles. The other document types were: book chapters (16), notes (7), short surveys (4), letters (3), conference reviews (2), books (2) and errata (2). The earliest record returned in this search on Scopus was an article by Price published in 1965 [15]. The number of outputs has gradually grown over the years, and the number of publications per year for the period 2006–2018 is shown in Figure 1 with the values listed in Table 1. The Compound Annual Growth Rate (CAGR) for years 2006 and 2016 for research outputs within the GSR domain was 8.0% ((last number/first number)^(1/periods)-1), a growth significantly higher than the annual change of 3.9% reported by the National Science Board, Science and Engineering Indicators 2018 [16]. The Average Annual Growth Rate (AAGR) (2006–2018) within the GSR domain was 11.2% with a standard deviation of 29.6%, Figure 1 and Table 1, revealing a steady growth despite some variability between years.

The country of origin of the 731 outputs was extracted from the authors' affiliations, and the results are shown in Figure 2. The United States of America had the largest paper count, followed by Brazil and Italy. For documents with authors from more than one country, each country was counted individually. The analysis does not provide a count related to the number of authors, but this option is available in the R code developed for this work [14].

**Table 1.** Scopus search results for the time period 2006–2018, covering 399 publications out of a total of 731. Table for all years (1965–2018) can be found in [14].

| Year | Publications [1] | | | | | | Authors | | | Ratio [4] | Author Percentage [5] |
|------|-------|------|------|-------|-------------|-------|-------|--------|---------------------|-----------|----------------------|
| | Total | Art. | Rev. | Conf. | Book Ch. | Other | Total | New [2] | First and Only [3] | | |
| 2006 | 19 | 13 | 2 | 4 | 0 | 0 | 72 | 47 | 33 | 3.8 | 65.3 |
| 2007 | 20 | 16 | 0 | 2 | 1 | 1 | 84 | 52 | 39 | 4.2 | 61.9 |
| 2008 | 12 | 8 | 0 | 2 | 2 | 0 | 41 | 21 | 17 | 3.4 | 51.2 |
| 2009 | 17 | 14 | 1 | 2 | 0 | 0 | 44 | 35 | 24 | 2.6 | 79.5 |
| 2010 | 20 | 14 | 1 | 5 | 0 | 0 | 75 | 59 | 43 | 3.8 | 78.7 |
| 2011 | 27 | 22 | 0 | 5 | 0 | 0 | 119 | 74 | 51 | 4.4 | 62.2 |
| 2012 | 45 | 34 | 1 | 2 | 7 | 1 | 187 | 127 | 73 | 4.2 | 67.9 |
| 2013 | 38 | 28 | 5 | 3 | 2 | 0 | 177 | 96 | 81 | 4.7 | 54.2 |
| 2014 | 37 | 28 | 2 | 6 | 1 | 0 | 139 | 80 | 62 | 3.8 | 57.6 |
| 2015 | 31 | 26 | 2 | 2 | 0 | 1 | 143 | 74 | 57 | 4.6 | 51.7 |
| 2016 | 41 | 33 | 2 | 2 | 3 | 1 | 178 | 89 | 76 | 4.3 | 50.0 |
| 2017 | 47 | 40 | 2 | 2 | 1 | 2 | 179 | 107 | 91 | 3.8 | 59.8 |
| 2018 | 45 | 36 | 6 | 1 | 0 | 2 | 194 | 113 | 110 | 4.3 | 58.2 |

[1] Conference paper and conference review are combined, idem for book and book chapter; [2] new Authors who had their first publication on gunshot residue in that year; [3] new Authors who had their first publication on gunshot residue in that year and have not contributed to other research output on gunshot residue since; [4] Ratio = Total number of Authors to Total number of publications; [5] percentage of New Authors to Total Authors.

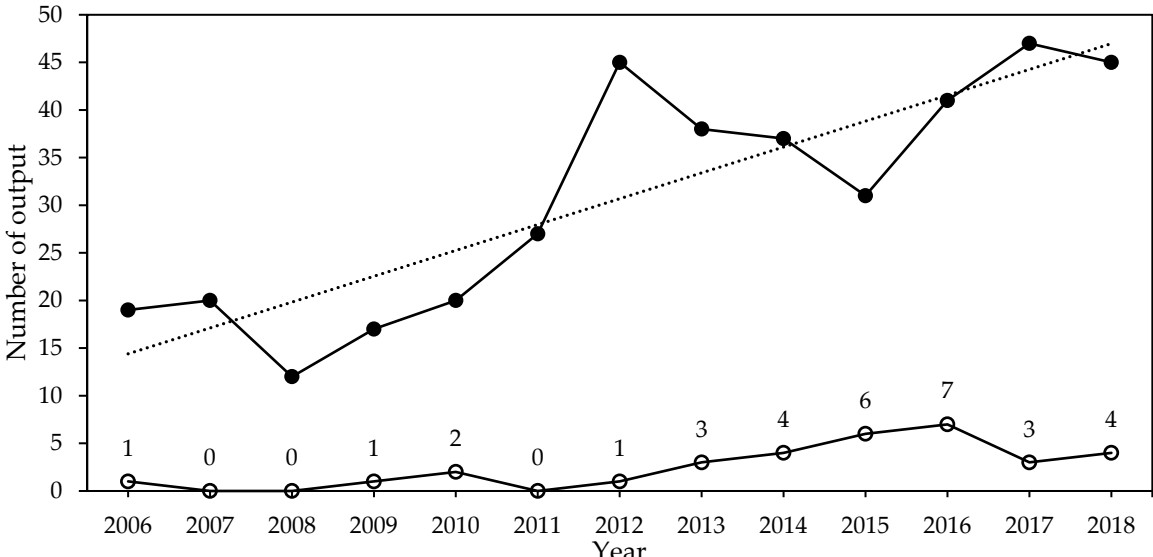

**Figure 1.** Scopus search for gunshot residue. ● indicates the total number of records for the year and ○ the number of Open Access documents.

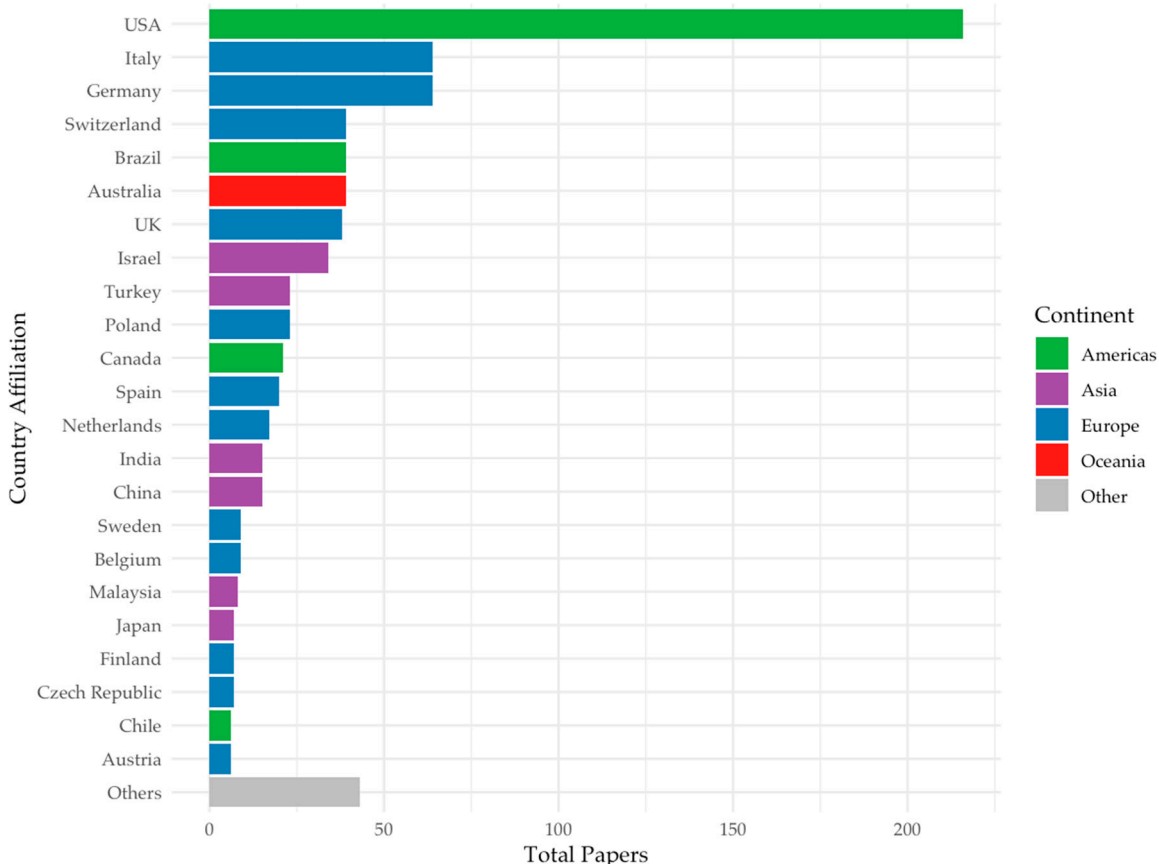

**Figure 2.** Country contribution found in literature on gunshot residue. The countries are colour-coded per geographical location. For outputs with authors from more than one country, a count was attributed to each individual country listed. Countries with <5 papers were collated in the 'Others' category (n = 25).

### 3.2. Open Access

Out of the 731 publications, only 35 (4.8%) were Open Access (OA), with the first OA article published in 1995 Ninomiya et al. on the use of Glazing Incidence X-Ray Fluorescence Analysis [17], followed by Zeichner and Glattstein's review in 2002 [18] and the article by Tuğcu et al. in 2005 [19]. The number of OA publications for all the subsequent years is shown in Figure 1. The year 2015 had the highest percentage of OA publications with 19.4% (6 out of 31), followed by 17.1% (7 out of 41) in 2016 and 10.8% (4 out of 37) in 2014; however, these percentages are well below the values reported by Piwowar et al. in their survey of the literature between 2009 and 2015, where they demonstrated that 27.9% of all journal articles with Crossref Digital Object Identifiers (DOIs), and 36.1% of all journal articles with Web of Science DOIs were Open Access between the years 2009 and 2015. While Piwowar et al. did not examine forensic science evidence types specifically, examining the subject areas (chemistry, physics, engineering and technology, etc.) used to classify GSR within the 731 outputs, revealed that 31.6% of publications associated with Physics, 15.5% associated with Chemistry and 17.4% associated with Engineering and Technology were Open Access, all far in excess of the 4.8% within the 731 publications reviewed.

### 3.3. Authors

The total number of authors between 2006 and 2018 was collated, Table 1, as well as the number of authors new to publishing in the field and authors who produced only one publication during the review period. The average number of authors per publication per year was 4, varying from a minimum ratio of 2.6 in 2009 to a maximum of 4.7 in 2013, but marginally increasing from an average ratio of 3.5 in the years between 2006 and 2009 to 4.2 in the years 2010 to 2018.

The number of new authors per publication (New Author Percentage) was found not to change significantly over the years, with an overall average of 61.4% (2006–2018). The percentage of those who only published once remained surprisingly constant across the study time frame, at 77% (2006–2018). One may have expected that new authors in early years would have had more chances of contributing to the research topic in subsequent years, but this was not observed. This may be because publications within the forensic science domain frequently involve practitioners working on case-specific scenarios rather than academic researchers working on more long-term research projects.

The Scopus research output of 731 on gunshot residue was generated by a total of 2533 authors (3 outputs had no authors listed), of whom 1589 were distinct and 401 contributed to two or more research publications, producing a total of 522 publications. The co-authorship network between the 401 authors is shown in Figure 3, where the nodes represent authors and the edges represent co-authorship. Eigenvectors were calculated for each node where the radius of the author's node is proportional to their eigenvector centrality (maximum value 1). A high eigenvector score signifies that a particular node is connected to many other nodes with high scores (this is a differentiation from in-degree centrality which refers instead to the number of receiving links by that node). The thickness of the edge and the line between two nodes is proportional to the number of interactions between the two nodes. The authors' eigenvector centrality reveals a long tail distribution where only two authors had an eigenvector centrality greater than 0.9 and 22 authors had an eigenvector centrality of greater than 0.5.

In Figure 3, not all the nodes are connected to one another, and there are four main clusters and 59 smaller groups of authors. The absence of bridging between authors means that the predominant authors published with members of their own research networks. The name of the authors who have their last publication in the years 2016 to 2018 are also indicated by colour coding. In red are those with their latest output on gunshot residue in 2018, green in 2017, and blue in 2016.

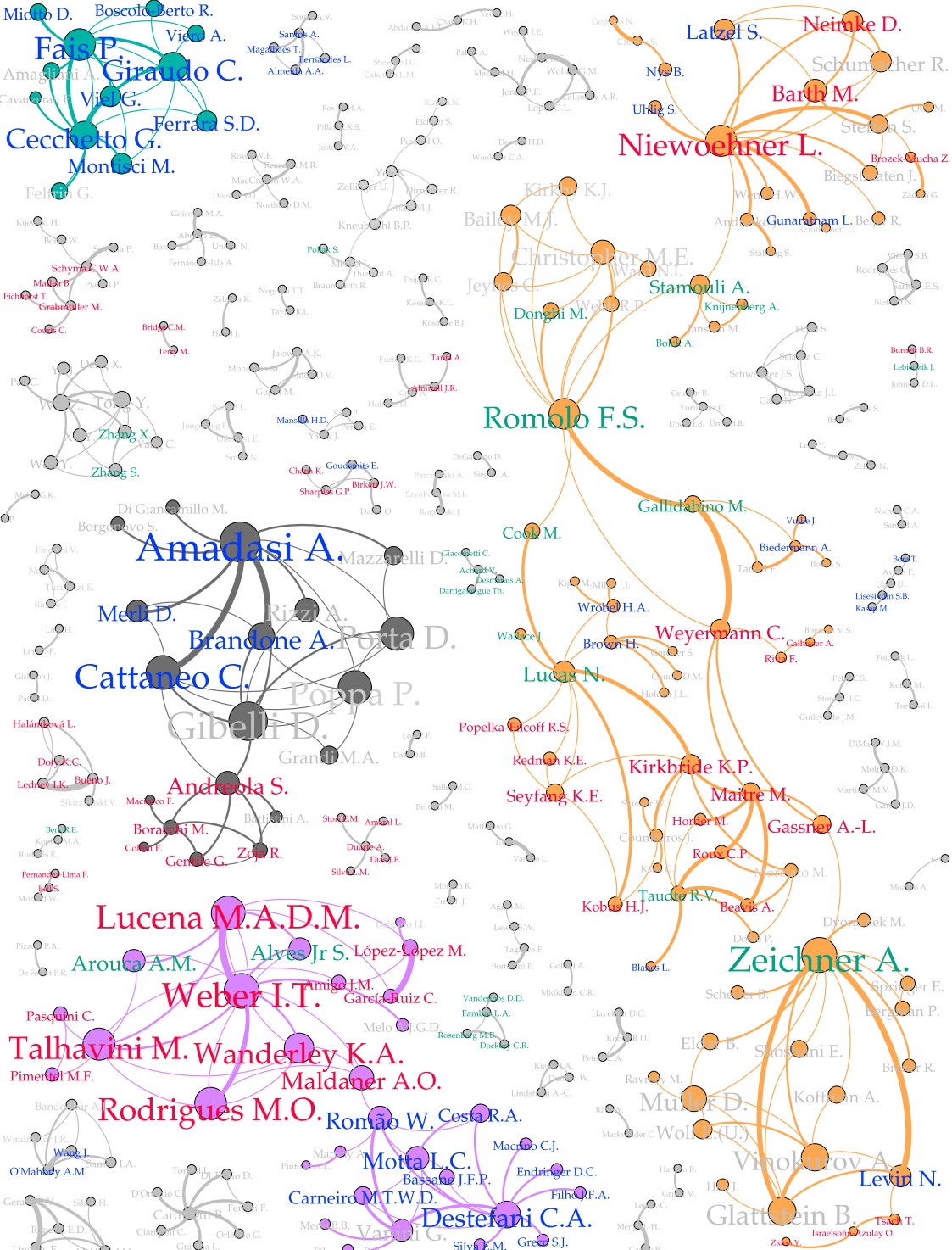

**Figure 3.** Co-authorship network generated by Gephi and ForceAtlas2 algorithm. Authors who have contributed to more than one publication on gunshot residue were considered, selected using the R code. The nodes represent authors and the edges co-authorship. Eigenvector centrality is used to determine the size of the node. The four largest networks (i.e., group of authors) are in colour. The width of the edge is proportional to the number of contributions between the two authors. The name of the authors who have their last publication in 2016 or later are colour-coded: 2016 in blue, 2017 in green and 2018 in red.

*3.4. Citations*

Scopus provides categorisation by discipline and includes a citation count for each of the outputs. Forensic Science is a multidisciplinary field, and outputs were attributed to more than one subject area. A total of 372 papers were classified within a single discipline with 247 classified as Medicine, 61 as Chemistry and 20 as Physics. Outputs were also associated across two or more disciplines, and those with the greatest numbers of publications were Biochemistry and Medicine with 134 publications, Medicine and Social Sciences with 64 publications and Computer Science, Engineering, Materials, Mathematics and Physics with 22 outputs each.

Review papers received high numbers of citations as expected [20,21] and are shown in Table 2. While it is a fairly straightforward metric, the number of citations is not, however, always a good indicator of research quality, and a publication can also receive a large number of citations for being controversial or reporting poor quality results. Older outputs are more likely to have a higher cumulative number of citations than more recent publications. When available from Scopus, the Field-Weighted Citation Impact (FWCI) is included for each of the records. This can be a more useful way to make a comparison between publications rather than just the number of citations alone, as they also take into account various factors, including the age of the different outputs. A score of 1.00 signifies that the publication received the expected number of citations, a number greater than 1.00 implies the article is doing better than expected, while a value less than 1.00 means the article is underperforming. With a FWCI score of 5.65, the review article by O. Dalby et al. [21] received almost 6 times the number of citations of an average article in the field, placing it as the best cited publication of 731 outputs in the Scopus list. However, while FWCI and the number of citations can help in comparing the performance of different publications, such an approach is not without controversy.

**Table 2.** Most cited publications; Scopus search all year to 2018: gunshot AND residue.

| Reference | Year | Authors | FWCI [1] | Citation | Scopus List [2] |
|:---:|:---:|:---:|:---:|:---:|:---:|
| [20] | 2001 | F.S. Romolo and P. Margo | 2.37 | 201 | 166 |
| [21] | 2010 | O. Dalby et al. | 5.65 | 173 | 137 |
| [22] | 1981 | K. Bratin et al. | NA | 148 | 19 |
| [23] | 2003 | M.J. Thali et al. | 1.70 | 123 | 5 |
| [24] | 1997 | H.-H. Meng and B.Caddy | 0.91 | 119 | 97 |
| [25] | 2001 | W. Thormann et al. | 2.14 | 89 | 3 |
| [26] | 1991 | D.M. Northrop et al. | NA | 85 | 19 |
| [27] | 1982 | S. Basu | NA | 75 | 65 |
| [28] | 2003 | A. Zeichner | 0.61 | 68 | 56 |
| [29] | 2009 | P.H.R. Ng et al. | 2.06 | 68 | 5 |

[1] Field-Weighted Citation Impact. Available from the Scopus database (1996–to present); [2] Publications citing the entry also present in the initial "gunshot AND residue" Scopus output list.

Figure 4 shows the distribution of the records for publications with a number of citations greater than 4 by the end of 2018. The mean number of citations per paper is 12, the median is 6 and the mode is 0. A total of 383 records were processed and grouped by subject area as a function of their normalised percentage citation provenance. For example, the review by Romolo and Margot [20], which is grouped into Medicine, had been cited 201 times, and 166 (82.6%) of these citations were from publications which were also included in the "gunshot AND residue" Scopus search output (i.e., the list of 731 records), from Table 2. With a percentage of 82.6%, the citation provenance of this work can be considered mostly internal to the original search criteria. Such an analysis requires each record to have a sufficient number of citations to begin with. For Medicine, 10 and 9 publications score 0% and 100%, respectively, as shown in Figure 4. Low numbers of citations (e.g., one or two) of some of these

publications can explain these scores; however, some of these outputs also have a reasonable number of citations. For example, the work by Blus et al. [30] has been externally cited 45 times. At the time of writing, out of the 731 records, 120 had no citations, and this may be because many publications were also recent (17 published in 2018, 9 in 2017 and 6 in 2016).

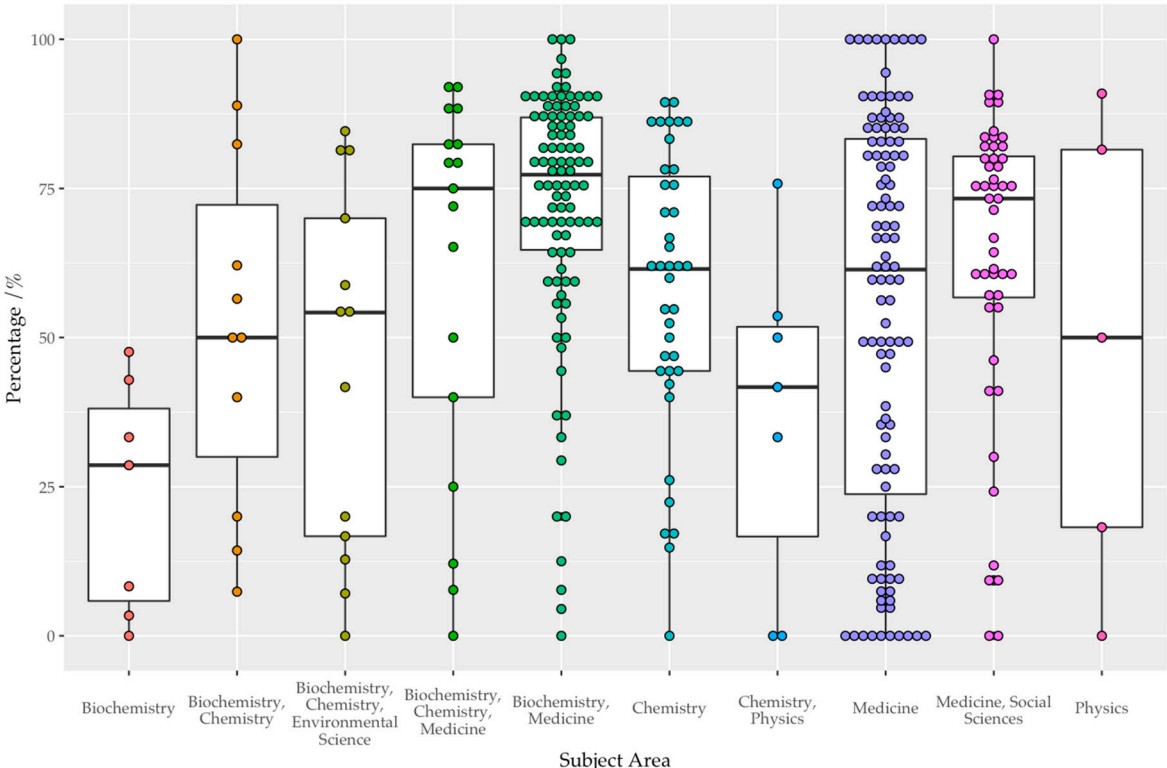

**Figure 4.** Percentage of citation per subject area already present with the original search. As of September 2019, only publications with 5 or more citations were selected (n = 413). Of the 413 records, 2 had been cited only in publications dating from 2018.

With the exception of Biochemistry, Chemistry and Physics and Physics, which have very small sample sizes, the box plots in Figure 4 are found to overlap, suggesting there is no significant difference between the different groups. The subject groupings of Biochemistry with Medicine and Medicine with Social Sciences were found to contain greater numbers of publications from within the GSR literature than other groupings, as defined by the Scopus search criteria. Citation provenances which are mostly internal to the original search criteria may indicate signs of an inward thinking research focus and community, reducing the wider impact of the research work.

## 3.5. Keywords

With Scopus, each record comes with two lists of keywords—the Author keywords and the Index keywords. The Author keywords are selected by the authors or journal editors as being considered relevant to the publication content. The Index keywords are produced by Scopus and generated by Elsevier's life science thesaurus Emtree®, which describes the publication's content. Such a list is specific to Scopus and will differ from other lists which could be generated by other databases such as Web of Science. In this analysis, only the results from the combined lists of keywords provided by Scopus is discussed, and no comparison was made to other databases.

From the 731 publications, 2716 Authors' keywords were used, of which 1433 are distinct, and 10,469 Index keywords were used, with 2880 being distinct. These keywords were combined together in order to generate a comprehensive list which was corrected for errors in spelling and harmonisation between acronyms and techniques. Understandingly, combining the Authors' and the

Index keywords from Scopus makes any subsequent analysis biased toward this specific database. Using only the Authors' keywords resulted in very few data points to be meaningfully interpreted. The year of first appearance and the frequency of the top 95 most commonly listed keywords are presented in Figure 5.

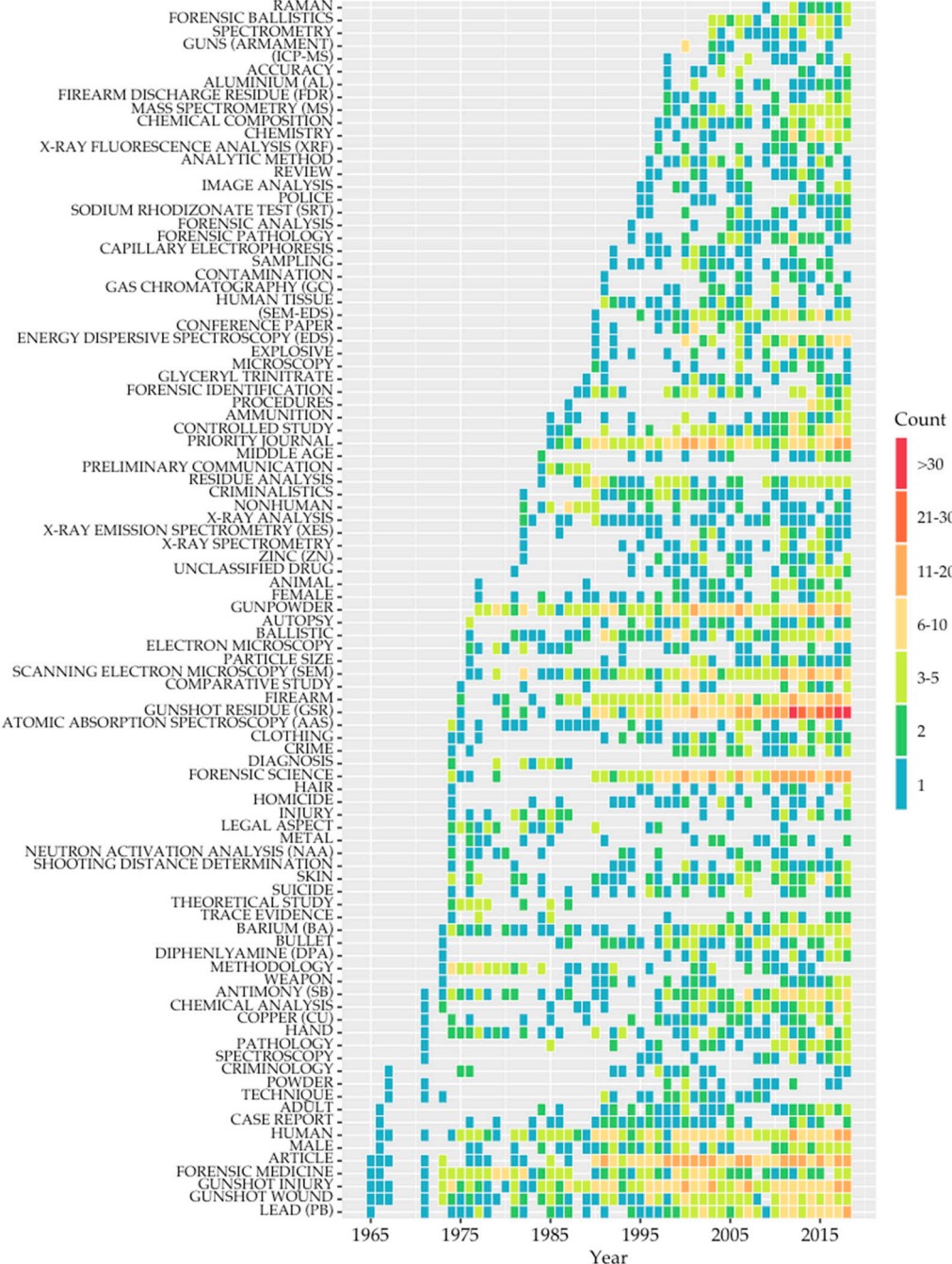

**Figure 5.** Most commonly found keywords in Scopus search for "gunshot AND residue". List ordered by first year of appearance.

Research activity and studies associated with gunshot residue in the various subject areas shown in Figure 4 and the broad spectrum of keywords observed in Figure 5 confirm this observation. Sorting keywords from this Scopus search by year of appearance reveals that GSR was first mentioned in 1975 by Goleb and Midkiff Jr [31]. This coincides with the works in the mid-1970s on the analysis of GSR by Keeley using scanning electron microscopy (SEM) and x-ray spectroscopy [32] and the detection of

gunshot residue using photoluminescence by Jones and Nesbitt [33]. The distribution of the keywords in Figure 5 confirms the historical trends in the study of GSR [20]. Until the mid-1970s, research effort was mostly focussed on using a bulk analysis approach, such as atomic adsorption spectroscopy (AAS) and neutron activation analysis (NAA). From the early 1980s, commercial SEM–energy-dispersive x-ray spectroscopy (EDS) systems became more readily available, and the detection and analysis of gunshot residue clearly benefited from improving instrument performance but also by capitalising on scientific advancement in other disciplines. To list a few from Figure 5, gas chromatography (GC), mass spectrometry (MS), inductively coupled plasma-mass spectrometry (ICP-MS) and more recently Raman are techniques used for the characterisation of gunshot residue. Some keywords fell out of fashion over the years, for example, "technique", "theoretical study", "legal aspect", "diagnosis" or "preliminary communication".

## 4. Discussion

In this study, a bibliometric analysis was performed in order to evaluate publication and citation trends in gunshot residue studies, the interconnectivity between authors, and keyword evolution. Electronic searches simplify this process, although they are limited to what is present within the database searched (i.e., Scopus), and a more comprehensive picture could be obtained by combining the output from multiple search engines. Corrections were also applied to the dataset so as to address spelling errors, acronyms and multiple names for similar techniques. Not all the search engines provide the same level of information, and careful consideration therefore needs to be taken if the results of search engines are to be combined. For example, in the author network analysis, Amadasi was found to be the most prolific in co-authorship, and their last paper on gunshot residue was published in 2016. Using the search engine ResearchGate, another paper on gunshot residue was published in 2018 by Amadasi et al. [34]. This discrepancy arose because ResearchGate reported the initial publication date as July 2018 and the *Journal of Forensic Science* where the article was published did so in 2019, and as such, the paper was not included within the Scopus search results. This is a common issue where articles are regularly published as "Advanced Access" or "Articles in Press" versions before being assigned to a specific volume and issue, meaning that the publication year can change from the "Articles in Press" version to the final version.

The methodology used in this form of analysis of the subject specific literature facilitated specific trends and analysis to be elucidated easily and systematically. The author network analysis revealed seven out of the top ten co-authors published in the last three years, demonstrating that gunshot residue continues to receive interest as a type of physical trace evidence. As seen in the keywords list, scanning electron microscopy remains the technique of choice for the detection and analysis of GSR, but thanks to improvements in analytical techniques and increased sensitivity, other scientific instruments are now also being considered to further expand our understanding of the physical and chemical properties of these particles.

The use of keyword analysis allowed certain trends across subjects to be highlighted, as well as the absence of specific areas where research remains to be conducted, such as studies in transfer and persistence, which were not well represented in the list of the most common keywords, being reported just four and ten times, respectively, across all 731 publications.

Nine out of the ten most cited publications included in the Scopus search presented here were published prior or simultaneously to the reports scrutinising forensic science, namely, the 2009 NAS report [2], the 2016 PCAST report [3] and the 2019 House of Lords Select Committee on Science and Technology report [4]. Only the publication by Dalby et al. "Analysis of gunshot residue and associated materials—A review" [21] was published the year following the release of the 2009 NAS report. From the ten most cited publications, none are Open Access. However, the selection of a publication should not rely only on citations as they may not all be directly relevant to the topic. One example is the article by Ng et al.; "Detection of illicit substances in fingerprints by infrared spectral imaging" [29]. Gunshot residue as the keyword is only mentioned twice throughout the publication by the authors stating that

Fourier-transform infrared spectroscopy (FTIR) and Raman spectral imaging can assist in the forensic analysis of gunshot residue.

The publications by Meng and Caddy [24], Romolo and Margo [20] and Dalby et al. [21] stand out as the three most cited reviews of the scientific literature on the identification and analysis of gunshot residue. Meng and Caddy focus on what gunshot residue is, how it is formed, collection techniques, common methods for analysis and how the results of these techniques can be interpreted on this type of evidence [24]. Romolo and Margo take on a more detailed approach, drawing attention to particle analysis and casework experience and introducing probabilistic methods to strengthen evidential value [20]. The authors do not discuss deposition, transfer or persistence of GSR but acknowledge that these matters are not well understood and further work is necessary to provide better interpretation of evidence. The more recent and extensive analysis by Dalby et al. [21] presents some of the challenges in regard to GSR as a trace evidence and discusses some of the factors influencing the analysis and interpretation of gunshot residue. Aspects considered are the distribution and transfer of GSR following a firearm discharge, the shooter's activities post-discharging the firearm and the subsequent effect on GSR in terms of loss. The environmental sources of GSR-like particles and contamination following an arrest are also mentioned by Dalby et al. [21].

The detection and the characterisation of GSR has been greatly helped with constant advancement in instrumental development. For example, starting in the 1980s, with higher computational power and better electronics, SEM instruments became more automated, making the task of searching for GRS particles on a sample less laborious. In a survey of 200 laboratories in the United States in 1988 (144 participating) by DeGaetano and Siegel, the finding of one particle having the appearance and elemental characteristics of gunshot residue for 41% of the respondents (i.e., 7 out of the 17 respondent to the question) was enough to indicate the presence of GSR [35]. The authors also reported there were some differences between the experts regarding the significance given to detecting one GSR particle, an observation later taken by Maitre et al. in their review of the current interpretation of gunshot residue and the application of Bayes theorem [36]. Interestingly, the study by DeGaetano and Siegel also discussed some of the issues encountered by the experts during the analysis of GSR with SEM–EDS, with one being the lengthy time required by the operator to carry out the analysis [35]. At the time, the authors suggested further studies should be carried out to evaluate the detection of GSR with SEM–EDS using automated methods. The findings of this survey were found to be largely supported by the results of another study carried out in 1992 by Singer et al. [37,38]. It was also found by that time that 83% of the laboratories polled were using automated SEM/EDS systems for GSR analysis [Singer1996], demonstrating forensic facilities' interest in employing new method to improve their workload. With manufacturers continuously adding new technology to their latest instruments to secure their position as market provider, their contribution in developing and implementing new methods has played an important part in facilitating the detection of GSR.

There are two more observations that are of interest from the two surveys. The first is an issue reported by DeGaetano and Siegel from the analysts' responses: "GSR is collected too long after the incident occurs" [35]. The second is regarding the literature: "Short of test firing comparative ammunition in the same type of weapon used in the crime, analysts cannot point to one or two references and produce the universal magic number for what constitutes a positive" [37]. The two remarks are unconnected yet clearly identify research need in regard to the transfer, persistence and background abundance of GSR.

Using the Scopus search and the keyword results, the keyword "background" only appears once in the keyword list, in the review by Maitre et al. [36]. There are a few more studies investigating the background frequency of GSR [39–41]. The keyword "frequency of occurrence" is also listed three times, once by Hannigan et al. [42] and twice by Brozek-Mucha et al. [43,44], whose studies involved surveying participants immediately after discharging a weapon. These studies are also complementary to other studies by the same authors investigating the persistence of GSR on hands [41,45]. Even though terms "background" and "frequency of occurrence" are mentioned here, it should be stated they are

not interchangeable; background level implies the amount of materials present in the environment, while frequency of occurrence is the amount of times a determined event occurs. "Transfer" as a keyword is mentioned eleven times in the Authors' keywords, either on its own or as part of a group of words (e.g., secondary transfer, energy transfer), and sixteen times when including the Index keywords. "Persistence", on the other hand, is listed thirteen times using both combined keyword lists. It is important to state the number of keyword occurrences is not assumed to be an accurate representation of the scientific content of all the listed outputs, and this information should be used as a guide, not as an alternative to reading the publications of interest. It is, however, concerning that when searching for terms such as "background" (and "frequency of occurrence"), "transfer" and "persistence" for a trace evidence such as gunshot residue, there are so few studies published given the critical importance of this type of information in the evaluation of the evidential weight of such evidence (a total of 33 individual output have keywords referring to background, transfer and/or persistence of GSR, and only two of them are openly accessible [46,47]).

The first Open Access paper of those reviewed was the 2014 study by Taudte et al. on the detection of GSR with mass spectrometry [48], and none of the most highly cited papers published from the mid-1990s (the time when Open Access first became available) were Open Access. One of the greatest challenges to accessing publication is the cost associated with it, and Open Access publications can substantially lower or remove the price barriers for the readers, while at the same time increasing the reach of the publication. However, the cost associated with Open Access publishing is borne by the individual authors or their organisations, and this may well be a significant barrier to publication through this route. While Open Access allows the results to reach a greater audience, on its own, it does not make research more transparent, and it is equally as important to make the data that underpin the research work also available where possible. The definition of data can vary significantly between disciplines, but they are not just extra materials placed in supplementary information. For gunshot residue analysis, for example, they could include the original SEM images in TIFF format, or the Raman, UV and mass spectrometry files in JCAMP format for ease of sharing and long term preservation [49]. These data should be placed in a repository with individual Digital Object Identifier (DOI) numbers and the reader guided to the dataset through the publication. Access to such data provides a means of validation of an experimental protocol or can be used as a reference for ongoing research, as well as a basis for new work to be generated, further improving the impact of the initial research [50,51]. At the time of submission (December 2019), none of the 731 outputs have data attached or provided reference to a DOI associated with the data, and the methodology used for the experimental studies performed was often omitted from the publications.

Gunshot residue is the combination of burned and unburned materials derived from the combustion of the primer that is formed following the discharge of a firearm. As mentioned before, thanks to technological advancement, the detection and characterisation of GSR particles is predominantly carried out using scanning electron microscopy combined with energy-dispersive x-ray spectroscopy (SEM–EDS), as seen in Figure 5. The combination of SEM with EDS provides both morphological information as well as qualitative elemental analysis. The results could also be quantitative. Most modern SEM–EDS instruments come with built-in reference libraries (for standardised and/or standardless quantitative analysis), but it is important to remember these reference data are generally obtained on standards and not on materials closely related to the sample of interest. In addition to this, for quantitative results to be reasonably accurate, a sample must be stable, flat, uniform in depth and free from voids. GSR particles have a stable morphology and elemental composition and are not affected by the necessary vacuum conditions of SEM–EDS, but GSR does not satisfy the other three requirements for accurate quantitative analysis. Gunshot residue particles are of an irregular and spherical shape. Under the spheroidal category, particles may be displayed as a perfect sphere but they also might appear stretched, dented or even distorted [52,53].

The detection and the characterisation of GSR has been associated with scanning electron microscopy since the 1970s and appropriately remains the preferred method for such investigation.

While the technique faces some challenges in providing quantitative analysis, this is not a limitation when it comes to the characterisation of a GSR particle for its description as forensic evidence in court. Qualitative microanalysis is sufficient to demonstrate the elements identified from their characteristic x-ray peaks where present, and their abundances can be left undetermined. To address this question, with advancement in scientific knowledge, research has started to look at other techniques that may be complementary to the SEM–EDS analysis in order to understand more about the properties and composition of GSR particles. Recent studies included a wide range of techniques, such as powder x-ray diffraction (XRD) [54], x-ray spectrometry by total reflection (TXRF) [55], solid-phase microextraction (SPME) [56,57] or focussed ion beam (FIB) [58,59]. Some of these techniques are included in the INTERPOL Review Papers 2019 [60], and further information about the advantages and limitations of the techniques can be found in relevant literature. With sufficient development and progress, some of these approaches may become part of the practitioners' toolkit, while others will only remain focused on fundamental research. The other aim of investigating the potential use of these techniques and methods is to address the growing availability of heavy metal free cartridges, even though without official figures being released from manufacturers, it is difficult to establish the current market share of lead-free ammunitions. SEM–EDS is facing some challenges in detecting particles from lead-free ammunition, but it is interesting to see studies have started to look at these heavy-metal free particles without any consideration of reference materials (see, for example, [56,61,62] and reference within). The detection limits of the techniques mentioned above are known from the information provided by the manufacturers. Results obtained from reference standards allow the customer or user to make an informed decision on which instrument will be best suited for their needs. The reference samples are, however, not always representative of the type of materials encountered in day to day applications, and the performance of an instrument may ultimately be affected (i.e., poorer resolution, detection limit, etc.). It is therefore important that analytical results from standard materials be acquired under the same experimental conditions as part of the data submission. An alternative would be to consider designed or commercially available samples more closely related to the type of materials collected by practitioners in order to be more relevant to casework.

It is interesting to notice there is no research on proxy materials as an alternative to GSR. This is supported by the keyword analysis where the keyword "proxy" does not appear in the keyword list. Not included in the output of the Scopus gunshot residue search, Meng and Lin [63] generated valuable research on the particle analysis of lighter flint residues using scanning electron microscopy/energy-dispersive X-ray spectrometry as the particles generated by flints used in the lighters were reported to resemble gunshot residue particles. The development of a proxy material would reduce the necessity to rely on a shooting range and on firearms for the supply of gunshot residue for standards and for research purposes, for example, in transfer and persistence experiments. Developing a proxy material is not without challenges, and a sufficient quantity of GSR particles must also be generated for comparative purposes. Closer connections to material-science-based research could facilitate such proxy development through already existing synthesis processes combined with investigations of the surface properties of the materials generated.

## 5. Conclusions

The use of a scientometric approach to the review of publications within a forensic science domain is novel. It provides a mechanism for identifying the key trends in the relevant literature and generating easy-to-understand graphical outputs which facilitate the visualisation of the publication landscape. Within the analysis of GSR, this review has highlighted gaps which need to be addressed in regard to the research effort, and these in particular relate to an understanding of the transfer, persistence and background abundance of GSR as a material. The development of characterised fit-for-purpose proxy materials for GSR will greatly facilitate the development of such research. The availability of research publications and their associated datasets through Open Access is also an issue that requires attention.

Such openness allows studies to be reproduced and addresses issues regarding the reliability of the results obtained and would be a progressive and positive step forward.

**Author Contributions:** Conceptualization, H.M. and N.N.D.; Methodology, C.S. and H.M.; Software (R-analysis), J.K.K., C.C. and H.M.; Writing—original draft preparation, C.S., J.K.K., C.C., N.N.D. and H.M. All authors have read and agreed to the published version of the manuscript.

**Funding:** This research was funded by the Leverhulme Trust grant RC-1015-011.

**Conflicts of Interest:** The authors declare no conflict of interest.

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
