# Peer review of "Reviewing Research Trends—A Scientometric Approach Using Gunshot Residue (GSR) Literature as an Example"

_publications, doi:10.3390/publications8010007_

Round 1

Reviewer 1 Report

This submission covers an interesting topic, especially since it is finally allowed to perform some types of gun-related research in the USA.

Some remarks

Line 39: open science is not a method, but an approach

I would suggest to draw one regression line in figure 1, showing the general trend. I do not think that there are enough data to support the finer analysis provided by the authors. Moreover, the authors do not have an explanation for the results of their finer analysis.

Reference 6 is written by the Open Science Collaboration. I think the editors of the journal should decide how to report this collective author.

Reference 53. Its editor is M.M. Houck.

Authors practice what they preach by making all data underpinning their research available.

Linguistic remarks

Line 17  I would write “a broad spectrum”

Line 51  Instead of “A better way of undertaking the scientific research must be where datasets are openly provided, associated to their publications”, I suggest “A better  way of undertaking scientific research happens (occurs?)  when, together with publications, related datasets are openly provided,”

Line 116 I do not think that authors are reviewed, but are collected.

Line 435 I would write “for comparative purposes”

Author Response

This submission covers an interesting topic, especially since it is finally allowed to perform some types of gun-related research in the USA.

We thank the reviewer for their constructive comments that has led to an improvement of the article.

Some remarks

Line 39: open science is not a method, but an approach

We agree with the reviewer. The wording has not been changed to: “Open science has been suggested as one approach to address the trends…”

I would suggest to draw one regression line in figure 1, showing the general trend. I do not think that there are enough data to support the finer analysis provided by the authors. Moreover, the authors do not have an explanation for the results of their finer analysis.

The reviewer is correct to point there is no need to have multiple linear regressions in Figure 1. They were only intended to complement the description found at the end of section 3.1. A general linear trend has been added to Figure 1 and the text was simplified to:

“…The Average Annual Growth Rate (AAGR) (2006-2018) within the GSR domain was 11.2 % with a standard deviation of 29.6 %, Figure 1 and Table 1, revealing a steady growth despite some variability between years.”

Reference 6 is written by the Open Science Collaboration. I think the editors of the journal should decide how to report this collective author.

The citation tool available in Science was used to download Reference 6. The Reviewer is correct to indicate in the publication list, reference 6: “Collaboration, O. S., Estimating the reproducibility…” should have read “Open Science Collaboration. Estimating the reproducibility…” This has now been corrected in the text.

Open Science Collaboration. Estimating the reproducibility of psychological science. Science 2015, 349.

We could not find in other articles citing Reference 6 a list of all the authors instead of Open Science Collaboration, but we will be happy to listen to the editors of the journal.

Reference 53. Its editor is M.M. Houck.

The reviewer is correct to indicate the reference is missing the Editor. The text in the revised manuscript: “Review Papers, 19th INTERPOL International Forensic Science Managers Symposium; Houck, M.M., Ed. 2019; pp. 1-861.”

Review Papers, 19th INTERPOL International Forensic Science Managers Symposium; Houck, M.M., Ed. 2019; pp. 1-861.

Authors practice what they preach by making all data underpinning their research available.

We thank the reviewer for this comment.

Linguistic remarks

The authors thank the reviewer for the suggestions. All the remarks below have now been included in the revised text.

Line 17  I would write “a broad spectrum”

Line 51  Instead of “A better way of undertaking the scientific research must be where datasets are openly provided, associated to their publications”, I suggest “A better  way of undertaking scientific research happens (occurs?)  when, together with publications, related datasets are openly provided,”

Line 116 I do not think that authors are reviewed, but are collected.

Line 435 I would write “for comparative purposes”

The manuscript has also been English language edited.

Reviewer 2 Report

Dear authors,

This paper presents a thorough bibliometric analysis of research published in GSR. 

The reviewed study is very interesting since very few investigations have been published with this type of methodologies.
Although, in order to offer an improved article, several revisions are requested:

It would be very interesting to provide references related to bibliometry, scientometrics, scientific communication, etc. in general terms. These types of references come from the field of communication and library science, where to offer the benefits of these methodologies.

As for the methodology, as indicated, the analyzes carried out are interesting, although they could have been extended to institutions, countries, keywords, etc.

To perform this comparison, GSR´s production could be analyzed in order to identify patterns; for example, authors ’countries of residence, institutions, co-authorship trends, and collaboration networks, number of citations articles received, etc.

On the other hand, some sections are offered that should be improved to make a better understanding:

In the section “3.1 overview of data” it indicates that the majority of the 731 outputs were research articles (606), 58 were conference papers and 31 were review articles. There would be another 36 documents to be determined. I would complete the analysis by offering at least the typology of them.

In the section “3.2 open Access” it says that only 35 of the publications are OA, 4.8%. This data in the summary represents 5%. You must normalize this percentage data.

Table 1 is interesting, but it is confusing. For example, you offer a total of 1632 authors (2006-2018) when you indicate in the summary that you work on 1589 known authors. The sum of publications is 399, when the total is 731, I do not see where this data is obtained.
I think the information should be improved and clarified.

The rest of the text is correct, the graphics and tables.
The discussion section is very extensive and informative.

Author Response

Dear authors,

This paper presents a thorough bibliometric analysis of research published in GSR.

The reviewed study is very interesting since very few investigations have been published with this type of methodologies.

We thank the reviewer for their constructive comments that has led to an improvement of the article.

Although, in order to offer an improved article, several revisions are requested:

It would be very interesting to provide references related to bibliometry, scientometrics, scientific communication, etc. in general terms. These types of references come from the field of communication and library science, where to offer the benefits of these methodologies.

We agree with the reviewer the introduction needed improving by including general references on the topic. The added text to the introduction now reads:

“Bibliometrics is a statistical and quantitative method used to analyse publications and visualise networks associated with a body of literature (e.g. [8,9]). Such an approach is being employed in systematic reviews and can potentially reveal unknown relations between different areas of research, and such findings have been demonstrated in literature-based discovery (e.g. [10-12]) or the study of citation impact indicator (e.g. [13]). In this work, a simplified scientometric approach (a sub-field of bibliometrics with a focus on analysing the scientific literature) has been applied to the available literature on gunshot residue (GSR) to identify patterns and trends in for example, in keyword, co-authorship, collaborative networks, country of origin and citations rates.”

de Solla Price, D.J. Networks of Scientific Papers. Science 1965, 149, 510, doi:10.1126/science.149.3683.510. Thilakaratne, M.; Falkner, K.; Atapattu, T. A systematic review on literature-based discovery workflow. PeerJ Computer Science 2019, 5, e235, doi:10.7717/peerj-cs.235. Chen, C.; Song, M. Visualizing a field of research: A methodology of systematic scientometric reviews. PLOS ONE 2019, 14, e0223994, doi:10.1371/journal.pone.0223994. Kostoff, R.N.; Boylan, R.; Simons, G.R. Disruptive technology roadmaps. Technological Forecasting and Social Change 2004, 71, 141-159, doi:10.1016/S0040-1625(03)00048-9. Waltman, L. A review of the literature on citation impact indicators. Journal of Informetrics 2016, 10, 365-391, doi:10.1016/j.joi.2016.02.007.

As for the methodology, as indicated, the analyzes carried out are interesting, although they could have been extended to institutions, countries, keywords, etc.

To perform this comparison, GSR´s production could be analyzed in order to identify patterns; for example, authors ’countries of residence, institutions, co-authorship trends, and collaboration networks, number of citations articles received, etc.

We thank the reviewer for these two comments and suggestions. The keywords, the number of citations per articles, the co-authorship trends and the collaboration networks were included in the manuscript and illustrated in Figures 3 – 5 (revised manuscript) and Table 2. We agree with the reviewer further analysis can be provided and as their suggestion the country of origin of the outputs would be interesting to give. This is now part of the manuscript, including a new figure, which can be found in Section 3.1. The added text reads:

“The country of origin of the 731 outputs was extracted from the authors’ affiliations with and the results are shown in Figure 2. The United States of America had the largest paper count followed by Brazil and Italy. For documents with authors from more than one country, each country was counted individually. The analysis does not provide a count related to the number of authors but this option is available in the R code developed for this work [14].”

Sobreira, C.; Klu, J.; Cole, C.; Nic Daéid, N.; Ménard, H. Reviewing research trends - a scientometric approach using gunshot residue (GSR) literature as an example - dataset. 2019; https://doi.org/10.5281/zenodo.3582799

On the other hand, some sections are offered that should be improved to make a better understanding:

In the section “3.1 overview of data” it indicates that the majority of the 731 outputs were research articles (606), 58 were conference papers and 31 were review articles. There would be another 36 documents to be determined. I would complete the analysis by offering at least the typology of them.

The reviewer is correct to point out all the document type should have been listed. The text now reads:

“The majority of the 731 outputs were research articles (606), 58 were conference papers and 31 were review articles. The other document types were: book chapters (16), notes (7), short surveys (4), letters (3), conference reviews (2), books (2) and errata (2). The earliest record returned in this search on Scopus was an article by Price published in 1965 [15]. The number of outputs has gradually grown over the years, and the number of publications per year for the period 2006-2018 is shown in Figure 1 with the values listed in Table 1.”

Price, G. Firearms Discharge Residues on Hands. Journal of the Forensic Science Society 1965, 5, 199-200.

In the section “3.2 open Access” it says that only 35 of the publications are OA, 4.8%. This data in the summary represents 5%. You must normalize this percentage data.

The percentage in the abstract was unintentionally rounded to 5 %. The text in the abstract now reads: “Out of the total number of publications only 35 (4.8 %) were found to be Open Access (OA)”, and all the other numbers in the publication have been checked.

Table 1 is interesting, but it is confusing. For example, you offer a total of 1632 authors (2006-2018) when you indicate in the summary that you work on 1589 known authors. The sum of publications is 399, when the total is 731, I do not see where this data is obtained.

The reviewer is correct to ask for clarification regarding the total number of authors listed in Table 1 but also in the abstract.

The 731 publications have a total number of 2533 known authors (3 publications have no author). These output were generated by 1589 individual authors and 401 of them have published two or more documents.

The text now reads:

“The 731 Scopus research output on gunshot residue was generated by a total of 2533 authors (3 outputs had no authors listed), of which 1589 were distinct and 401 contributed to two or more research publications, producing a total of 522 publications.”

I think the information should be improved and clarified.

The rest of the text is correct, the graphics and tables.

The discussion section is very extensive and informative.

The manuscript has also been English language edited.